# Ferroptosis Inducers Upregulate PD-L1 in Recurrent Triple-Negative Breast Cancer

**DOI:** 10.3390/cancers16010155

**Published:** 2023-12-28

**Authors:** Christophe Desterke, Yao Xiang, Rima Elhage, Clémence Duruel, Yunhua Chang, Ahmed Hamaï

**Affiliations:** 1UFR Médecine-INSERM UMRS1310, Université Paris-Saclay, F-94800 Villejuif, France; 2INSERM UMR-S1151, CNRS UMR-S8253, Institut Necker Enfants Malades, Université Paris Cité, F-75015 Paris, France; yao.xiang@inserm.fr (Y.X.); rima.elhage@inserm.fr (R.E.); clemence.duruel@inserm.fr (C.D.); yunhua.chang-marchand@inserm.fr (Y.C.); 3Team 5/Ferostem Group, F-75015 Paris, France

**Keywords:** CD274, ferroptosis, immunotherapy, breast cancer, basal, TNBC

## Abstract

**Simple Summary:**

Triple-negative breast cancer (TNBC) is characterized by a quick and high rate of recurrence. The benefits from neo-adjuvant chemotherapy associated with anti-PDL1 have been shown to be efficient in this aggressive form of breast cancer. Ferroptosis inducers (erastin/RSL3) induced the upregulation of CD274 in TNBC cells. Basal and TNBC subtypes of breast cancers overexpressed CD274 conjointly with three ferroptosis drivers: TNFAIP3, IFNG and IDO1 (IDO1: inhibitory immune checkpoint). These tumors present higher levels of recurrence. A potential synergy of ferroptosis inducers with anti-PD-L1 immunotherapy is suggested in recurrent TNBC.

**Abstract:**

(1) Background: Triple-negative breast cancer (TNBC) is a distinct subgroup of breast cancer presenting a high level of recurrence, and neo-adjuvant chemotherapy is beneficial in its therapy management. Anti-PD-L1 immunotherapy improves the effect of neo-adjuvant therapy in TNBC. (2) Methods: Immune-modulation and ferroptosis-related R-packages were developed for integrative omics analyses under ferroptosis-inducer treatments: TNBC cells stimulated with ferroptosis inducers (GSE173905, GSE154425), single cell data (GSE191246) and mass spectrometry on breast cancer stem cells. Clinical association analyses were carried out with breast tumors (TCGA and METABRIC cohorts). Protein-level validation was investigated through protein atlas proteome experiments. (3) Results: Erastin/RSL3 ferroptosis inducers upregulate CD274 in TNBC cells (MDA-MB-231 and HCC38). In breast cancer, CD274 expression is associated with overall survival. Breast tumors presenting high expression of CD274 upregulated some ferroptosis drivers associated with prognosis: IDO1, IFNG and TNFAIP3. At the protein level, the induction of Cd274 and Tnfaip3 was confirmed in breast cancer stem cells under salinomycin treatment. In a 4T1 tumor treated with cyclophosphamide, the single cell expression of Cd274 was found to increase both in myeloid- and lymphoid-infiltrated cells, independently of its receptor Pdcd1. The CD274 ferroptosis-driver score computed on a breast tumor transcriptome stratified patients on their prognosis: low score was observed in the basal subgroup, with a higher level of recurrent risk scores (oncotypeDx, ggi and gene70 scores). In the METABRIC cohort, CD274, IDO1, IFNG and TNFAIP3 were found to be overexpressed in the TNBC subgroup. The CD274 ferroptosis-driver score was found to be associated with overall survival, independently of TNM classification and age diagnosis. The tumor expression of CD274, TNFAIP3, IFNG and IDO1, in a biopsy of breast ductal carcinoma, was confirmed at the protein level (4) Conclusions: Ferroptosis inducers upregulate PD-L1 in TNBC cells, known to be an effective target of immunotherapy in high-risk early TNBC patients who received neo-adjuvant therapy. Basal and TNBC tumors highly expressed CD274 and ferroptosis drivers: IFNG, TNFAIP3 and IDO1. The CD274 ferroptosis-driver score is associated with prognosis and to the risk of recurrence in breast cancer. A potential synergy of ferroptosis inducers with anti-PD-L1 immunotherapy is suggested for recurrent TNBC.

## 1. Introduction

Breast cancer is a highly heterogenous disease, classified in distinct subgroups based on transcriptomic profiles that are associated with outcome [1]. Expression experiments initially identified four breast cancer ‘intrinsic’ subtypes (basal-like, HER2-enriched, luminal and normal breast-like) [2], and subsequent studies have led to the sub-stratification of luminal breast cancers into luminal A and luminal B and shown that this classification system is of prognostic significance [3].

The overlap ratio can be as high as 60–90% between basal-like breast cancer (BLBC) and triple-negative breast cancer (without expression of hormonal receptors) (TNBC), compared to only 11.5% between non-TNBC and BLBC [2,4]. High-risk early triple-negative breast cancer is frequently associated with early recurrence and high mortality [5]. In this subgroup of aggressive breast cancer, neo-adjuvant chemotherapy is the preferred treatment approach [6]. Immune checkpoint inhibition may enhance endogenous anticancer immunity after the increased release of tumor-specific antigens with chemotherapy [7]. Pembrolizumab is an anti-programmed death 1 (PD-1) monoclonal antibody. Among patients with early triple-negative breast cancer, the percentage with a pathological complete response was significantly higher among those who received pembrolizumab plus neo-adjuvant chemotherapy than among those who received placebo plus neo-adjuvant chemotherapy [8].

Ferroptosis is non-apoptosis-regulating cell death. Unlike autophagy and apoptosis, ferroptosis is an iron-dependent and reactive oxygen species (ROS)-reliant cell death, with characteristics mainly of cytological changes, including decreased or vanished mitochondria cristae, a ruptured outer mitochondrial membrane and a condensed mitochondrial membrane [9]. The majority of TNBC patients presented somatic genetic alterations affecting the TP53 gene: up to 80 percent in the cancer subgroup [10]. p53 is a transcription factor, which plays its tumor-suppressive role through selective transcriptional regulation of many target genes implicated in cellular responses, like apoptosis, cell cycle arrest, DNA repair and metabolism [11]. Emerging evidence shows that ferroptosis plays a crucial role in the tumor suppression of p53. p53 functions as a key bidirectional regulator of ferroptosis by adjusting the metabolism of iron, lipids, glutathione peroxidase 4, reactive oxygen species and amino acids via a canonical pathway [12].

Accumulating evidence supports that ferroptosis regulates tumor progression by releasing multiple signal molecules in the tumor microenvironment and plays a key role in cancer biology and drug resistance [13]. Expression correlation between PD-L1 and ferroptosis-related genes has already been observed in the transcriptome of TNBC, especially a strong positive correlation with ACSL4 expression [14].

In the present work, we identify that erastin and RSL3 ferroptosis inducers upregulated CD274 in TNBC cells (MDA-MB-231 and HCC38). In breast tumors from the TCGA cohort, the expression of CD274 is associated with overall survival, and tumors presenting high expression levels of CD274 upregulated some ferroptosis-driver genes also associated with prognosis, like IDO1, IFNG and TNFAIP3. The CD274 ferroptosis-driver score computed on the breast tumor transcriptome stratified patients on their prognosis: low score was observed in the basal subgroup, presenting a higher level of recurrent risk scores such as oncotypeDx risk, genomic grade index (ggi) and gene70 score. In cohort METABRIC, CD274, IDO1, IFNG and TNFAIP3 were confirmed to be overexpressed in the TNBC subgroup. The CD274 ferroptosis-driver score was confirmed as an independent parameter for the overall survival of the patients according to the inclusion of age at diagnosis and Tumor, Nodes and Metastasis stage classification in the multivariate model. The tumoral expression of CD274, TNFAIP3, IFNG and IDO1 in biopsy of breast ductal carcinoma was confirmed at the protein level via immunohistochemistry imaging. 

## 2. Materials and Methods

### 2.1. Datasets of Cell Line Models Stimulated with Ferroptosis Inducers

#### 2.1.1. MDA-MB-231 RNA-Sequencing Transcriptome

RNA-sequencing FPKM quantification performed on MDA-MB-231 triple-negative breast cancer cell lines from dataset GSE173905 [15] was downloaded on Gene Expression Omnibus website [16] at the following web address: https://www.ncbi.nlm.nih.gov/geo/query/acc.cgi?acc=GSE173905 (accessed on 22 November 2023). During these triplicate experiments, MDA-MB-231 cell line was treated over 72 h with two distinct ferroptosis inducers: erastin and RSL3.

#### 2.1.2. HCC38 Microarray Transcriptome

Normalized matrix of microarray from dataset GSE154425 [17] was downloaded at the following web address: https://www.ncbi.nlm.nih.gov/geo/query/acc.cgi?acc=GSE154425 (accessed on 22 November 2023) and annotated with the corresponding technology platform: GPL17692, Affymetrix Human Gene 2.1 ST Array, available at the web address: https://www.ncbi.nlm.nih.gov/geo/query/acc.cgi?acc=GPL17692 (accessed on 22 November 2023). These triplicate experiments were performed on HCC38 triple-negative breast cancer cell line. Experimental conditions of this transcriptome dataset comprised vehicle control and in vitro stimulations over 18 h with erastin (ferroptosis inducer), with tubacin (HDAC6 inhibitor) and with combination of the two drugs.

#### 2.1.3. Single Cell Transcriptome for Immune Microenvironment of 4T1-Transplanted Tumors

Raw single cell transcriptome data (GEX, 10× genomics) of immune cells from 4T1 transplanted tumors with or without cyclophosphamide pre-treatment from dataset GSE191246 [18] were downloaded on Gene Expression Ominibus website [16].

#### 2.1.4. Proteomic Validation of PDL1 Regulation under Ferroptosis Inducer

Normalized mass-spectrometric data performed on breast CSC model (human mammary epithelial HMLER CD24low/CD44high) with salinomycin treatment were collected to test the regulation of PDL1 (CD274), PDCD1 (PD1), IDO, IFNG and TNFAIP3 [19].

### 2.2. Cohorts of Breast Cancer Tumor Samples

#### 2.2.1. Training Cohort: TCGA Firehose Invasive Breast Cancer Cohort

TCGA invasive breast cancer was examined [20] on Cbioportal [21]. RNA-sequencing transcriptome data were normalized by voom transformation in edgeR R-package version 3.38.4 [22,23]. Genefu R-package version 2.28.0 [24] was used to predict the three molecular classifications: pam50 subtypes [3,25], scmod2 subtypes [26], claudin-low characterized by low expression of tight junction proteins claudins 3, 4 and 7 and E-cadherin [27]. Genefu R-package was used to compute the three distinct predictive scores: oncotype, Dx, gene70 prediction [28], genomic grade index (ggi) [29] prediction. These scoring metadata were added to phenotype data of the samples in an experiment set (eset) also containing normalized RNA-sequencing data. An R-package entitled “tcga.breast” was compiled under version 1.0.0 with inclusion of this eset data [30].

#### 2.2.2. Validation Cohort: METABRIC Breast Cancer Cohort

METABRIC breast cancer transcriptome cohort [31,32,33] was investigated as validation cohort through the website Cbioportal [16]. This interactive data portal allowed us to perform oncoprint expression representation for CD274 and retained ferroptosis-driver genes. Relapse-free survival analysis was performed according to stratification transcriptome alterations observed for CD274 and retained ferroptosis-driver genes. 

### 2.3. Transcriptome Immune Modulation Scoring

Bioinformatics development and analyses were performed in R software environment 4.2.1. Based on gene lists implicated in immune landscape of cancer [34], an R-package was built to perform single sample expression enrichment on classes and subclasses of molecules identified as immune modulators [35] in transcriptome experiments. This package is named “immunemod” (version 1.0.0) and is available at the following web address: https://github.com/cdesterke/immunemod (accessed on 22 November 2023) [36].

### 2.4. Ferroptosis Visualization in Transcriptome Enrichment 

On TCGA breast cancer cohort, CD274 (PD-L1) expression was used to stratify the cohort in two groups (patients CD274-low and patients CD274-high) according to overall survival of the patients. This stratification was obtained with survminer R-package version 0.49, and survival was fitted with survival R-package version 3.5.7. Differential expression gene analysis between CD274-low and CD274-high patient samples was conducted with LIMMA R-package version 3.52.4 with thresholds on log2 fold change superior to 0.5 and on False-Discovery Rate (FDR)-adjusted *p*-value inferior to 0.05 [37]. Based on ferrDb V2 database [38], an R-package was built to improve visualization of ferroptosis-related genes among differential expressed genes and is accessible at the address: https://github.com/cdesterke/ferroviz (accessed on 10 December 2023) [39].

### 2.5. CD274 Ferroptosis-Driver Gene Score Quantification

Loops of univariate overall survival analyses against expression of immune marker (CD274) and ferroptosis drivers were computed with “loopcolcox” R-package (version 1.0.1) available at the following web address: https://github.com/cdesterke/loopcolcox, accessed on 22 November 2023). For each significant marker (univariate Cox *p*-value ≤ 0.1), Cox beta coefficients were extracted to compute an expression score comprising CD274 and 3 ferroptosis drivers: TNFAIP3, IFNG and IDO1. This score is based on the following equation:$$score=\sum \left(exp*beta.coef\right)$$

On TCGA breast cancer transcriptome cohort, the exact equation of the score was: (Expression(IFNG) × −0.127036521465874) + (Expression(IDO1) × −0.0711938750267523) + (Expression(TNFAIP3) × −0.152997397587969) + (Expression(CD274) × −0.102709239674522). Optimal score threshold stratification according to overall survival of the patients was obtained with survminer R-package version 0.49, and survival model was fitted with survival R-package version 3.5.7. Transcriptome expression heatmap and associated unsupervised clustering (Euclidean distances) were drawn with pheatmap R-package version 1.0.12. Multivariate overall survival model was constructed with survival R-package version 3.5.7 by including CD274 ferroptosis score categories with age at diagnosis and Tumor, Nodes, Metastasis (TNM) classification of the tumors. The multivariate overall survival model was assessed by performing individual and global Schoenfeld test. Univariate binomial analyses on clinical data were performed with Publish R-package version 2023.01.17. Graphical representations (boxplots and scatterplots) were drawn with ggplot2 R-package version 3.4.4 [40].

### 2.6. Immunohistochemistry

For markers employed to compute the CD274 ferroptosis-driver score (CD274, IDO1, IFNG and TNFAIP3), their respective protein expression in human breast tumor was verified by immunohistochemistry on samples processed by protein-atlas consortium [41].

### 2.7. Single Cell Transcriptome Analysis

Raw 10× Genomics data were used as input files to create individual Seurat object with Seurat R-package version 4.2.1 [42]. Single cell quality preprocessing was carried out via filtration of cell expression less than 300 features and via filtration of features expressed in less than 100 cells by experimental condition. Individual Seurat objects were integrated by canonical correlation and normalized on their common anchors. Data of the integrated object were scaled. A first-dimension reduction in the data was performed by principal component analysis (PCA) on thirty dimensions and a second one by UMAP on the twenty first dimensions of the PCA. Cells were clustered with graph-based algorithm [43] and visualized by dimplot, featuresplot and violinplot graphical representations.

### 2.8. Flow Cytometry

Expression of CD274 was tested by flow cytometry on human breast cancer stem cell model HMLER CD44^high^CD24^low^ but also on TNBC cell line MDA-MB-231 after 48 h treatments with distinct ferroptosis inducers: erastin, RSL3 and salinomycin. Briefly, 3 × 10^5^ cells were stained with a directly coupled anti-human CD274/PDL1-APC antibody (1:20; ref. 329708, BioLegend^®^, London, UK) in DPBS supplemented with 10% FBS during 30 min at 4 °C. After washing twice with DPBS, cells were suspended in the medium with 2.5 μg/mL DAPI (D3171, Invitrogen, Waltham, MA USA) to stain the dead cells. For each sample, 100,000 events were analyzed using a BD LSRFortessa flow cytometer (BD Biosciences, San Jose, CA, USA) and processed using BD FACSDiva software (BD Biosciences) https://www.bdbiosciences.com/en-us/products/software/instrument-software/bd-facsdiva-software, accessed on 22 December 2023, and FlowJo software https://www.flowjo.com/, accessed on 22 December 2023 (FLOWJO, LLC, Ashland, OR, USA).

## 3. Results

During this work, we performed integration omics experiments (bulk transcriptome, single cell RNAseq, mass spectrometry) performed on breast cancer cell models (MDA-MB-231, HCC38, 4T1, breast cancer stem cells) stimulated by distinct ferroptosis inducers (erastin, RSL3, salinomycin, cyclophosphamide). Conception of two R-packages “immunmod” (omics analysis of immune modulation molecules) and “ferroviz” help us to understand the relation between immunology and ferroptosis of the tumors. The associations between CD274 and ferroptosis-driver expressions were verified to be relevant in the clinic by expression score computing in two independent cohorts of breast cancer (TCGA and METABRIC). All these processes are summarized in the following workflow (Figure 1).

### 3.1. CD274 (PD-L1) Is Up-Regulated by Ferroptosis Inducers in Triple-Negative Breast Cancer Cells

The triple-negative breast cancer MDA-MB-231 cell line was stimulated with ferroptosis inducers (erastin, RSL3) over 72 h, and RNA sequencing was processed in these experimental conditions [12]. The immune modulation signature extracted from the cancer immune landscape [29] was investigated in these experiments. Single sample immune scoring showed potential immune regulation in the presence of ferroptosis inducers: erastin and RSL3 as compared to the vehicle control (Figure 2A). The RSL3 ferroptosis inducer showed a more important immune regulation than erastin (Figure 2A). A common increase in “inhibitory” immune checkpoints was observed for erastin and RSL3 stimulation as compared to control (Figure 2B), and it appeared to increase significantly with the “ligand subcategory” for both ferroptosis inducers (Figure 2C). The increase in the immune ligand score appeared more pronounced for RSL3 than for erastin stimulation (Figure 2B). These results suggest that ferroptosis inducers could increase the inhibitory immune checkpoint score in the transcriptome of triple-negative breast cancer cells.

The expression of inhibitory immune checkpoints was investigated in the same transcriptome dataset, and both ferroptosis inducers were shown to induce the significant overexpression of CD274, TGFB1 and ADORA2A (Figure 3A,B), with an especially marked overexpression of CD274 (Figure 3B,C). The expression of immune receptors [34] in MDA-MB-231 stimulated by erastin and RSL3 was also investigated. These analyses did not show any major significant regulation of immune receptors in MDA-MB-231 after stimulation with ferroptosis inducers (Appendix A).

CD274 expression was investigated in an independent dataset of transcriptome GSE154425 [17]. These experiments were carried out on an HCC38 triple-negative breast cancer cell line with stimulation over eighteen hours by erastin (ferroptosis inducer), tubacin (HDAC6 inhibitor) and combination of the two molecules (combo) (Figure 3D). These independent experiments confirmed that erastin could induce the overexpression of CD274 in the HCC38 TNBC cell line. These results suggest that ferroptosis inducer (erastin, RSL3) stimulation on TNBC cells increased the expression of CD274, a known target of cancer immunotherapy.

### 3.2. Breast Tumors with High Expression of CD274 Harbored Upregulation of Ferroptosis Drivers

The expression of CD274 was investigated in the transcriptome of breast tumors according to clinical-associated data of the TCGA invasive breast cohort. This transcriptome cohort composed of 1082 cases was stratified in two subgroups of samples according to CD274 expression and overall survival of the patients (Figure 4A). Patients with CD274 low expression (*n* = 426) in their tumor harbored a significantly shorter overall survival than patients with CD274 high expression (*n* = 656) in their tumor (log rank test *p*-value = 0.022, Figure 4B). These two groups of patients did not show significant differences in terms of age at diagnosis (*p*-value = 0.1871019, Table 1) or in terms of TNM classification (Table 1). In term of molecular classification, a significant difference was found between CD274-low and CD274-high patients (*p*-value = 0.0002416, Table 1), with a higher proportion of basal and Her2 subgroups in CD274-high patients. Among the predictive score, a significant difference was found for the G70 score (*p*-value = 0.0005523, Table 1) but nothing significant for OncotypeDx and ggi scores (Table 1). Differential expression gene analysis highlighted a significant increase in the expression for ferroptosis drivers in CD274-high samples, as compared to CD274-low samples (Figure 4C). These results confirm a high connection between ferroptosis induction and CD274 expression in breast cancer. By mass spectrometry analysis performed in a breast cancer stem cell model (human mammary epithelial HMLER CD24low/CD44high) [19], salinomycin, a ferroptosis inducer [44], was capable of inducing protein upregulation of CD274 and TNFAIP3 (Figure 4D,E) without any modulation of PDCD1. The upregulation of CD274 at the surface protein level was confirmed by flow cytometry analyses in MDA-MB-231 (TNBC cell line) under treatment with erastin and RSL3 (Figure 3E) but also in a breast cancer stem cell model (human mammary epithelial HMLER CD24low/CD44high) under treatment with erastin, RSL3 and salinomycin (Figure 4F).

### 3.3. CD274 Ferroptosis-Driver Score Predicts Recurrence of Breast Cancer

In the breast tumor transcriptome from the TCGA cohort, univariate overall survival analyses according to the expression of ferroptosis drivers highlighted a significant favorable expression for three of them: IFNG, IDO1 and TNFAIP3 (Figure 5A) conjointly with CD274 expression (univariate Cox *p*-value = 0.08). A CD274 ferroptosis -driver score combining the expression and Cox beta-coefficients of CD274, IFNG, TNFAIP3 and IDO1 was computed. According to the overall survival of the patients, an optimal threshold cutoff point was determined in order to stratify the cohort in two groups in relation to this score (Figure 5B): patients with low score (*n* = 153) and patients with high score (*n* = 929). Log rank test confirmed a significant difference in terms of overall survival between these two groups of patients, with longer survival time for patients in the low-score category (log rank test *p*-value = 0.0042, Figure 5C). In terms of TNM classification, no significant difference was observed in tumor stage (*p*-value = 0.06, Table 2) and metastasis stage (*p*-value = 0.29, Table 2), but a significantly higher proportion of node stage 2 and 3 was observed in the high-score group (*p*-value = 0.006, Table 2). Unsupervised clustering performed on the expression of the four markers confirmed the score stratification of patients with a left cluster enriched in patients with low score and high expression of the four molecules (Figure 5D). This left cluster of low-score patients tends to be of the basal type with fewer events of death and presenting an ontypeDX risk (Figure 5D). Concerning pam50 molecular classification, a significantly higher proportion of basal and Her2+ was observed in the low-score group (*p*-value < 0.0001, Table 2).

Cyclophosphamide is a neo-adjuvant chemotherapy used with success in association with pembrolizumab in metaplastic triple-negative breast cancer [45], but cyclophosphamide is also known as a ferroptosis inducer because cyclophosphamide-induced GPX4 degradation triggers parthanatos by activating AIFM1 [46] and also acts on Heme Oxygenase-1 [47]. The single cell transcriptome of the microenvironment infiltrating in 4T1-transplanted tumors was investigated with or without cyclophosphamide pre-treatment [18]. In the tumor microenvironment generated by 4T1 transplantation in mice, the infiltration of lymphoid (T and B) and myeloid (Mo/Macrophage) and dendritic cells could be observed (Figure 6A), and these cell subpopulations are quantitative variables after cyclophosphoshamide treatment (Figure 6B). For example, a reduction in B lymphocyte (red cluster 1) and an increase in T lymphocyte (blue cluster 0) infiltration was observed under cyclophosphamide treatment (Figure 6B). The phenotype of the cells is not dramatically affected under treatment because cell clusters are well structured according to experimental conditions after integration (Figure 6C). These single cell transcriptome analyses confirmed the induction of CD274 overexpression by cyclophophamide in the 4T1 tumor immune microenvironment, especially in B cells, in Monocyte/Macrophage Lyz2+ or Skfn4+ and in dendritic cells of Slamf7+ (Figure 6D–F). In the 4T1 tumor microenvironment, TNFAIP3 appeared well expressed in T and B lymphoids and the myeloid compartment, but after cyclophophamide stimulation, this ferroptosis driver was found upregulated in Mono/Macrophages skfn4+ and dendritic cells of Slamf7+ (Figure 6D and Appendix A). In these experiments on the 4T1 tumor microenvironment, Ido1 could not be detected. Pdcd1 (pd1) and Ifng were found invariably expressed in LyT Cd3e+-S100A10+ (Figure 6D, and Appendix A).

In the TCGA transcriptome cohort, patients with a lower CD274 ferroptosis-driver score were confirmed to be in the basal and Her2 subtypes (Figure 7A). In the basal and Her2 subgroups, a high expression was observed for CD274, IFNG and IDO1 (Appendix A). A high expression of TNFAIP3 was observed only in basal and normal-like subgroups (Appendix A). Concerning the SCMOD2 classifier, a low level of CD274 ferroptosis-driver score was observed in the ER-/Her2- group (Figure 7B). Interestingly, this CD274 ferroptosis-driver score was found significantly associated to distinct prognostic scores. Patients with low score harbored higher values of Genomic Grade Index (ggi) (*p*-value < 2.2 × 10^−16^, Figure 7C and Table 2), higher values of the OncotypeDx score (*p*-value < 2.2 × 10^−16^, Figure 7D, Appendix A and Table 2) and higher values of the gene70 score (*p*-value < 0.0001, Table 2).

The expression of CD274, IFNG, TNFAIP3 and IDO1 was investigated in an independent cohort of the transcriptome: METABRIC cohort comprising 1866 breast tumors. The majority of transcriptional alterations observed for these markers concerned upregulation, with a frequency superior or equal to 2 percent of the tumors (Figure 8A). Relapse-free survival patients presenting combined transcriptional alterations for these four markers presented a significantly longer survival time during the follow-up (log rank test *p* = 0.0127, Figure 8B). In terms of tumor grade for this validation cohort, a significant increase in tumor stage 3 was found associated in the group of patients affected by transcriptional alterations in these four genes (*p*-value < 10 × 10^−10^, Figure 8C). Patients affected by transcriptional alterations in these four genes presented a significant increase in basal and claudin-low subgroups (*p*-value < 10 × 10^−10^, Figure 8D). A higher expression of CD274 was found in the TNBC subgroup of the METABRIC cohort, as compared to other breast cancer samples (*p*-value = 1.25 × 10^−10^, Appendix A). A higher expression of IDO1 was found in the TNBC subgroup of the METABRIC cohort, as compared to other breast cancer samples (*p*-value < 2.2 × 10^−16^, Appendix A). A higher expression of IFNG was found in the TNBC subgroup of the METABRIC cohort, as compared to other breast cancer samples (*p*-value < 2.2 × 10^−16^, Appendix A). A higher expression of TNFAIP3 was found in the TNBC subgroup of the METABRIC cohort, as compared to other breast cancer samples (*p*-value < 2.2 × 10^−16^, Appendix A).

### 3.4. CD274 Ferroptosis-Driver Score Is an Independent Prognosis Marker in Breast Cancer Overall Survival

The independence of the CD274 ferroptosis-driver score in the breast cancer prognosis was investigated through building a multivariable overall survival model on the TCGA transcriptome cohort. This Cox multivariable model was performed with the inclusion of score stratification (Table 2), age at diagnosis stratification (threshold 65 years old) and TNM stage classification. The Global Schoenfeld test was used to assess the linearity of the residuals in the model at the multivariable level (Global Schoenfeld test, *p*-value = 0.07, Appendix A). The concordance index of the overall multivariate model reached a Ci equal to 0.722 (with standard error = 0.025). In the multivariate overall survival model, patients quantified with a high value of the CD274 ferroptosis-driver score harbored a significant positive hazard ratio (2.012 ± 1.063, multivariable *p*-value = 3.18 × 10^−2^, Table 3, Figure 9A). These results suggest/support that the CD274 ferroptosis-driver score is an adverse independent prognosis marker of the overall survival in breast cancer. At the protein level, via immunohistochemistry, a weak expression of CD274 and moderate expression of ferroptosis drivers (IFNG, TNFAIP3 and IDO1) were observed in biopsy of ductal breast carcinoma (Figure 9B).

## 4. Discussion

In the present work, we observed that ferroptosis inducers such as erastin and RSL3 increased the expression of PD-L1 in distinct triple-negative breast cancer cells. Anti-PD-L1 immunotherapy has been shown to improve pathological complete response of high-risk early triple-negative breast cancer patients who received neo-adjuvant chemotherapy [8].

Some synergy between the induction of ferroptosis and immunogenic cell death (ICD) has already been described to potentiate cancer anti-PD-L1 immunotherapy. For example, a tannic acid (TA)-Fe3+-coated doxorubicin (DOX)-encapsulated 1,2-distearoyl-sn-glycero-3-phosphoethanolamine-N-[methoxy(poly(ethylene glycol))-2000] (ammonium salt) (DSPE-PEG) micelle (TFDD) for apoptosis/ferroptosis-mediated immunogenic cell death (ICD) has been reported. By coating TA-Fe3+ on the surface of DOX-loaded micelles, an apoptotic agent and a ferroptotic agent are simultaneously delivered into the cancer cells and induce cell death. Furthermore, the intracellular oxidative environment generated by the apoptosis/ferroptosis hybrid pathway stimulates the endoplasmic reticulum (ER) and leads to ICD induction. The associated in vivo results show that the combination treatment of TFDD and anti-programmed death-ligand 1 antibodies (anti-PD-L1) considerably inhibits tumor growth and improves antitumor immunity by activating CD4+ and CD8+ T cells and decreasing the ratio of regulatory T cells (Treg) to CD4+ T cells [48]. The expression of PD-L1 has also been described as being positively correlated to the expression of ferroptosis-related genes such ACSL4 [14]. In breast tumor, we observed that breast tumors, which harbored high levels of CD274, upregulated the expression of ferroptosis-driver genes. Among these upregulated ferroptosis-driver genes, some of them were also found associated in their expression to the prognosis of the patients: IFNG, TNFAIP3 and IDO1. In the breast tumor transcriptome, the CD274 ferroptosis-driver score was found associated to the prognosis of breast cancer patients and also to the risk of recurrence (g70, ggi and oncotypeDx scores) [49]. Interferon gamma (IFNγ) released from CD8+ T cells downregulates the expression of SLC3A2 and SLC7A11, two subunits of the glutamate-cystine antiporter system xc-, impairs the uptake of cystine by tumour cells and, as a consequence, promotes tumor cell lipid peroxidation and ferroptosis [50]. A20/TNFAIP3 is significantly upregulated in TNBC, and its expression level is highly correlated with low/poor survival of metastasis-free patients, promoting cancer metastasis via multi-monoubiquitylation, which activates the functions of Snail [51]. In the BV-2 microglial cell line, knock-out of TNFAIP3 (A20) attenuates cell susceptibility to OGD/R-induced ferroptosis and upregulates inflammatory responses [52]. TNFAIP3, also known as A20, is a ubiquitin-editing enzyme and functions as an endogenous suppressor of NF-κB, which activates inflammation (Priem, van Loo and Bertrand, 2020). A20 restricts NF-κB signals through its deubiquitinase activity. A20 is regulated by microRNA (miRNA) and acts as a regulator of endothelial cell ferroptosis [53], and the A20/TNFAIP3-CDC20-CASP1 axis promotes inflammation-mediated metastatic disease in triple-negative breast cancer [54]. Indoleamine 2,3-dioxygenase 1 (IDO1) is an immunosuppressive enzyme involved in tumor immune escape. Blockade of the IDO1 pathway is an emerging modality of cancer immunotherapy. IDO1 expression in basal-like TNBCs is considered as an immune inhibitory signal that counterbalances active immunity and may reflect the high mutational load of these tumors [55]. IDO1 oxidizes tryptophan (TRP) to generate kynurenine (KYN), and KYN serves as the source for molecules that inhibit ferroptotic cell death [56].

## 5. Conclusions

Ferroptosis inducers upregulated PD-L1 in TNBC cells, known as an effective target of immunotherapy in high-risk early TNBC patients who received neo-adjuvant therapy. Basal and TNBC tumors highly expressed CD274 and ferroptosis drivers: IFNG, TNFAIP3 and IDO1. The CD274 ferroptosis-driver score is associated with the prognosis and risk of recurrence in breast cancer. A potential synergy of ferroptosis inducers with anti-PD-L1 immunotherapy is suggested in recurrent TNBC.

## Figures and Tables

**Figure 1 cancers-16-00155-f001:**
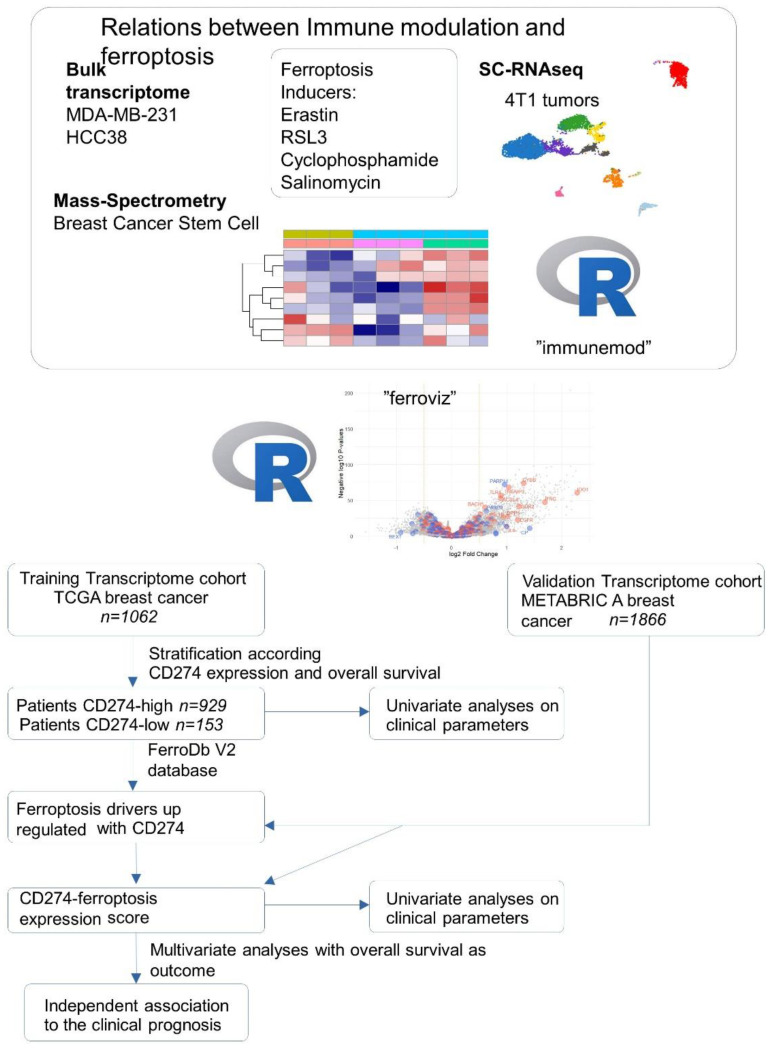
Workflow of data integration process during this study.

**Figure 2 cancers-16-00155-f002:**
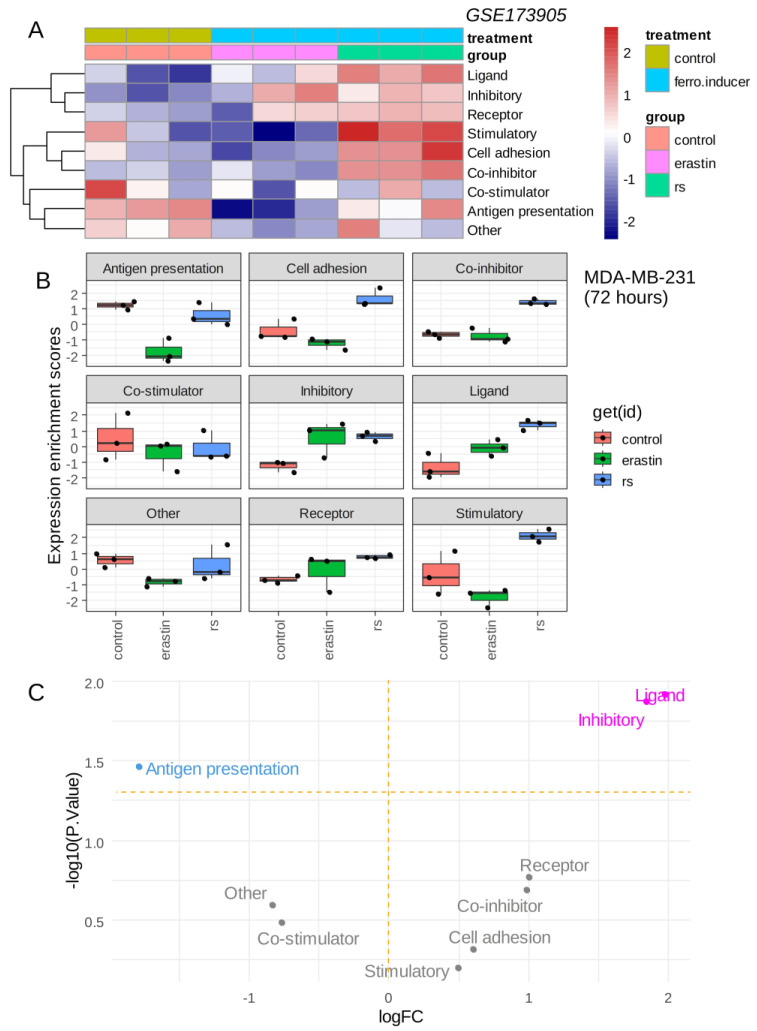
Ferroptosis inducers up-regulated inhibitory immune checkpoints in triple negative breast cancer cells: (**A**) Heatmap of immune modulation scores computed on transcriptome of MDA-MB-231 cells stimulated during 72 h with ferroptosis inducers: erastin and RSL3 (rs); (**B**) Boxplot of immune modulation scores computed on transcriptome of MDA-MB-231 cells stimulated during 72 h with ferroptosis inducers: erastin and RSL3 (rs); (**C**) Volcanoplot testing regulation of immune modulation scores computed on transcriptome of MDA-MB-231 cells stimulated or not during 72 h with ferroptosis inducers.

**Figure 3 cancers-16-00155-f003:**
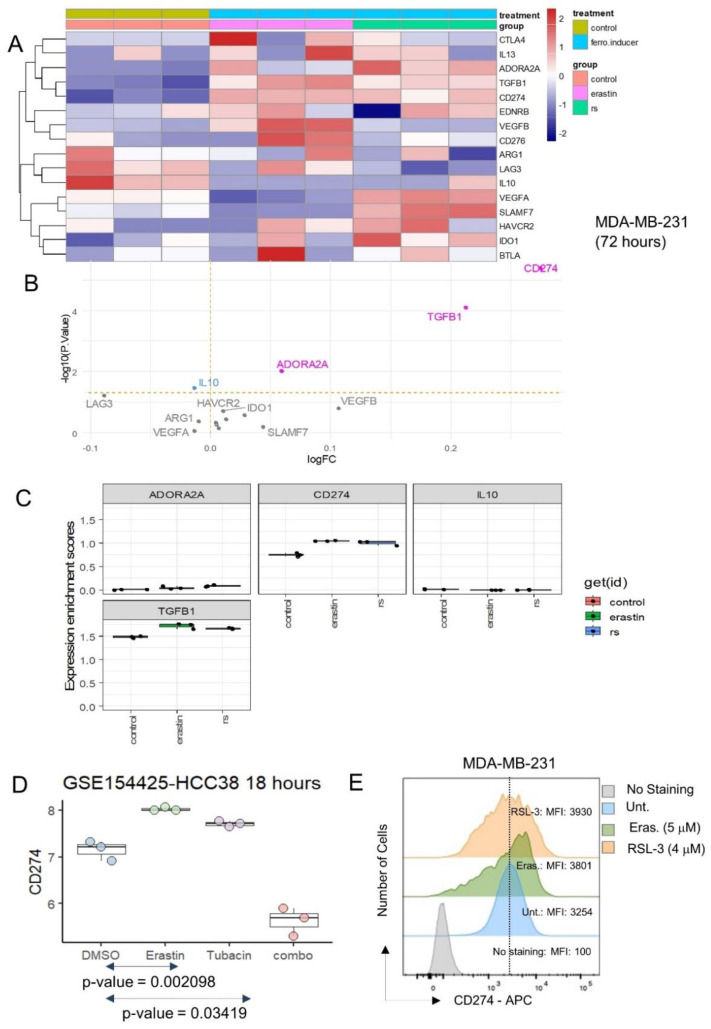
Ferroptosis inducers up-regulated PD-L1 (CD274) inhibitory immune checkpoint in triple negative breast cancer cells: (**A**) Expression heatmap of inhibitory immune checkpoints in transcriptome of MDA-MB-231 stimulated or not with ferroptosis inducers: erastin and RSL3 (rs); (**B**) Volcanoplot testing regulation of inhibitory immune checkpoints in MDA-MB-231 stimulated or not with ferroptosis inducers; (**C**) Boxplot of expression for inhibitory immune checkpoints in transcriptome of MDA-MB-231 stimulated or not with ferroptosis inducers: erastin and RSL3 (rs); (**D**) Expression boxplot CD274 in HCC38 triple negative breast cancer cells according stimulation with erastin ferroptosis inducer, tubacin (HDAC6 inhibitor) and combo (combination of erastin and tubacin); (**E**) Expression of CD274 by flow cytometry analysis in MDA-MB-231 under ferroptosis inducer treatments (48 h): erastin, and RSL3. MFI: mean of fluorescence intensity.

**Figure 4 cancers-16-00155-f004:**
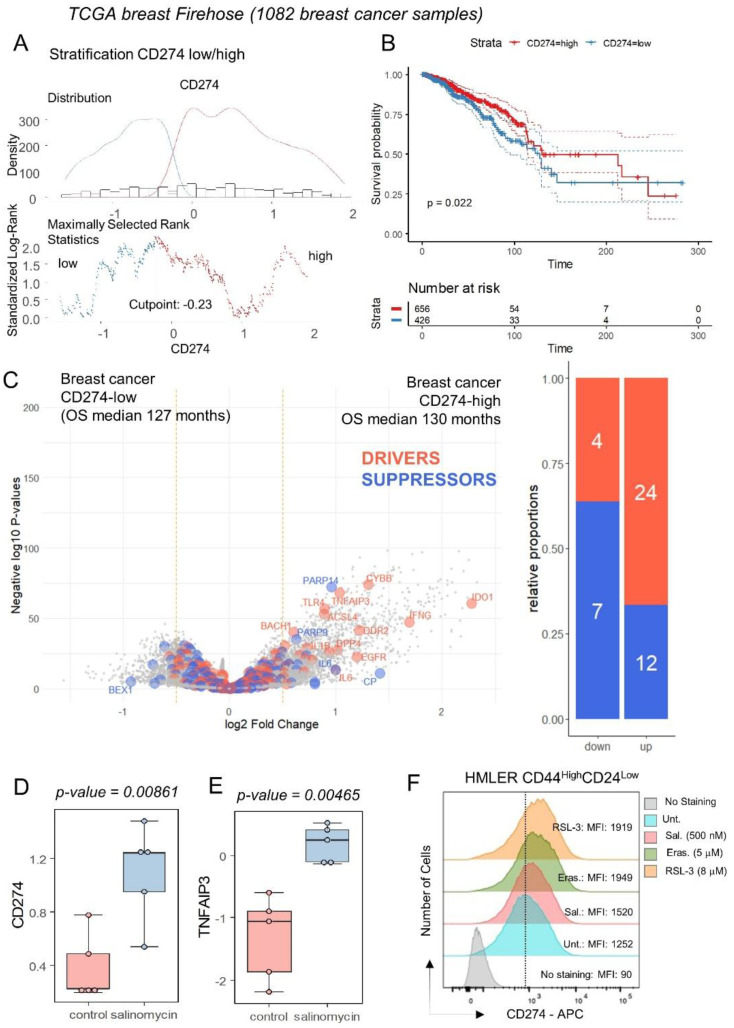
Breast tumors of patients with longer survival conjointly overexpressed CD274 and ferroptosis drivers: (**A**) Optimal threshold for CD274 expression according to overall survival of the patient from Firehose TCGA invasive breast cancer transcriptome cohort (*n* = 1082 tumors); (**B**) Kaplan Meier and log rank test for breast cancer TCGA cohort according to overall survival outcome and stratification on CD274 expression groups; (**C**) Volcanoplot and barplot of ferroptosis actors (FerrDbV2: drivers in orange and suppressor in blue) regulated between breast tumors according to their low or high level of CD274 expression; (**D**) Protein expression of CD274 by breast cancer stem cell under salinomycin stimulation; (**E**) Protein expression of TNFAIP3 by breast cancer stem cell under salinomycin stimulation. (**F**) Expression of CD274 by flow cytometry analysis in HMLER CD44^high^CD24^low^ under ferroptosis inducer treatments (48 h): erastin, RSL3, and salinomycin. MFI: mean of fluorescence intensity.

**Figure 5 cancers-16-00155-f005:**
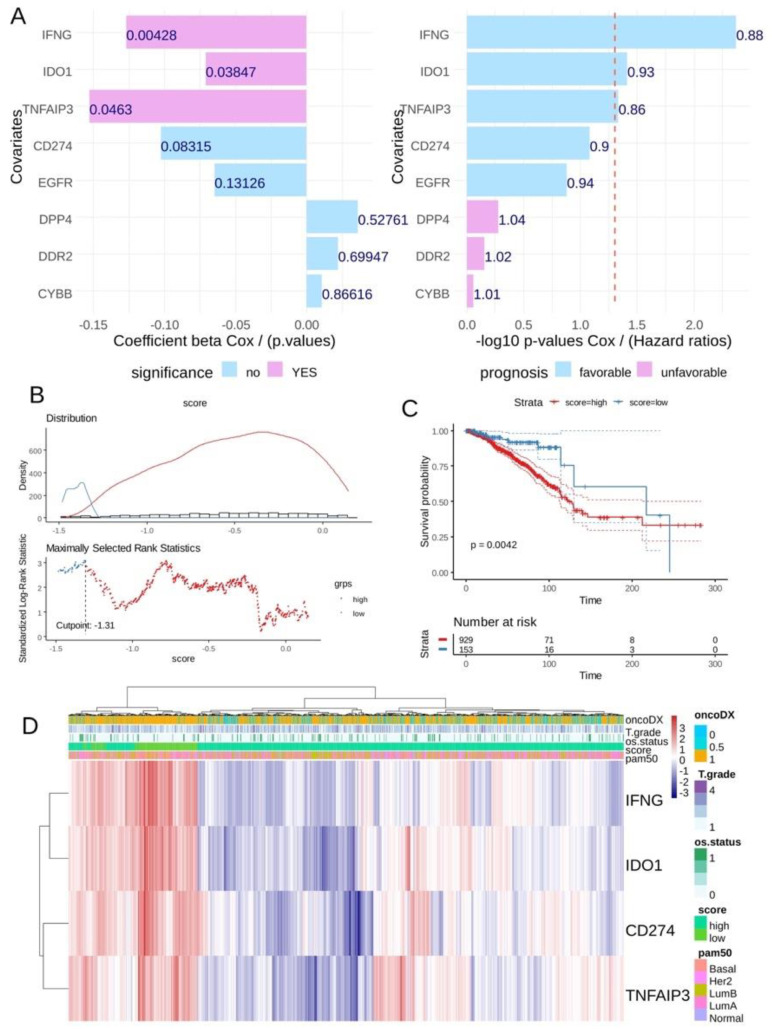
The CD274 ferroptosis score allowed us to stratify breast tumors according to their prognosis in TCGA training cohort: (**A**) Barplot of univariate Cox (beta coefficients and Negative log10 *p*-values) analysis for CD274 and ferroptosis drivers expression according to overall survival of patients in TCGA breast cancer cohort; (**B**) Optimal threshold determination for CD274-ferroptosis score according to overall survival of the patient (TCGA cohort); (**C**) Kaplan Meier and log rank test on overall survival (TCGA cohort) according to CD274-ferroptosis score group stratification; (**D**) Unsupervised clustering (Euclidean distances) and expression heatmap of CD274 and ferroptosis drivers used to compute expression score.

**Figure 6 cancers-16-00155-f006:**
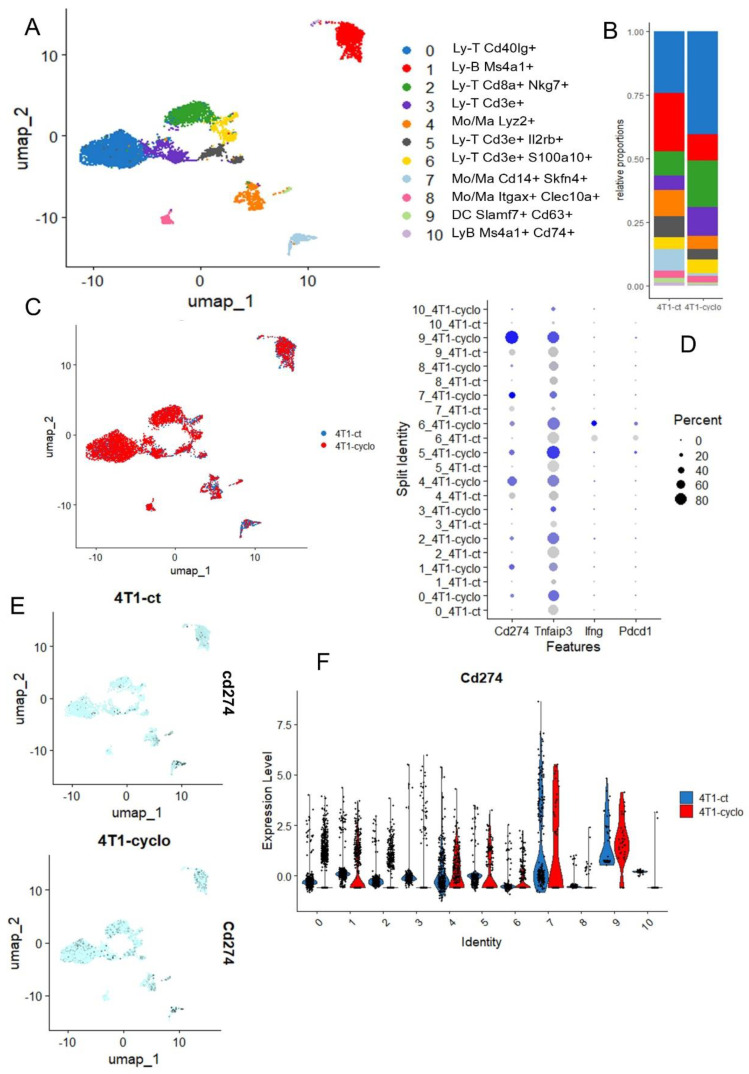
Single cell transcriptome of infiltrated microenvironment in 4T1 transplanted tumor after cyclophosphamide treatment: (**A**) UMAP dimension reduction with cell cluster identification; (**B**) Cell cluster proportion according to experimental conditions; (**C**) UMAP dimension reduction stratified on experimental conditions: CT (untreated control), CYCLO (cyclophosphamide treatment); (**D**) Dotplot of single cell expression for Cd274, Tnfaip3, Ifng and Pdcd1 across the distinct cell cluster and experimental conditions, color scale from grey to blue: level of expression; (**E**) Featureplot of Cd274 expression stratified on experimental conditions; (**F**) Violinplot of Cd274 expression across cell clusters and split on experimental conditions.

**Figure 7 cancers-16-00155-f007:**
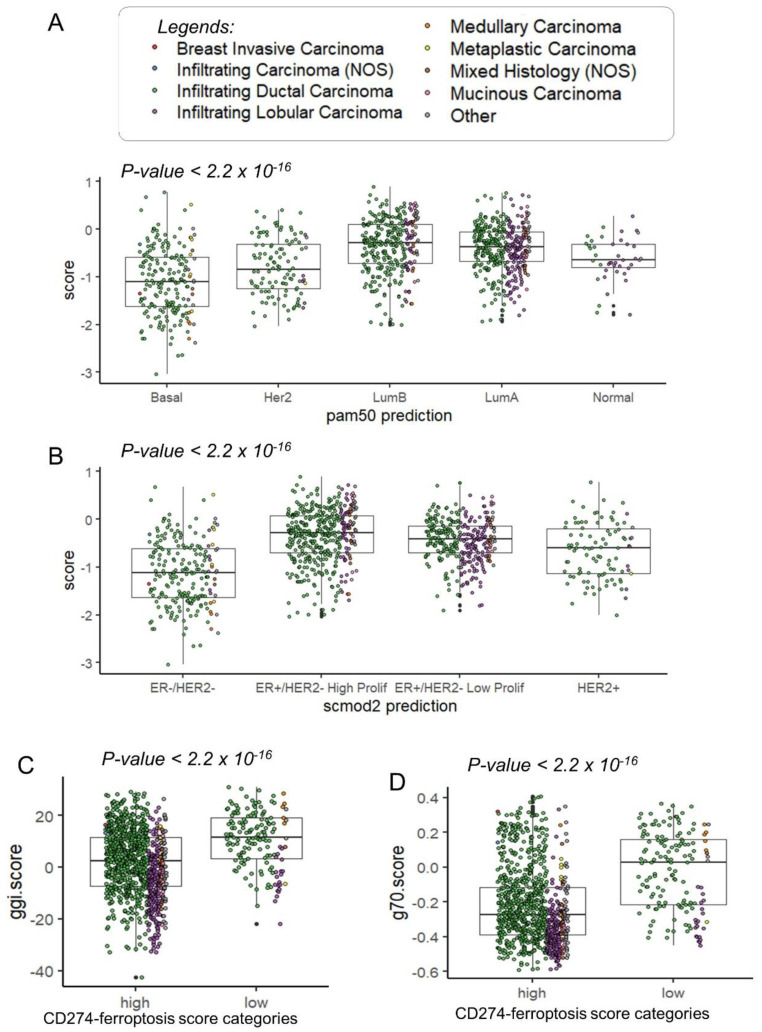
Clinical associations of CD274-ferroptosis driver score in TCGA breast cancer cohort: (**A**) Boxplot of CD274-ferroptosis driver score stratified on pam50 molecular classification (Fisher ANOVA *p*-value); (**B**) Boxplot of CD274-ferroptosis driver score stratified on scmod2 molecular classification (Fisher ANOVA *p*-value); (**C**) Boxplot of ggi molecular score stratified on categories of CD274-ferroptosis driver score (Student *t* test *p*-value); (**D**) Boxplot of ggi molecular score stratified on categories of CD274-ferroptosis driver score (Student *t* test *p*-value).

**Figure 8 cancers-16-00155-f008:**
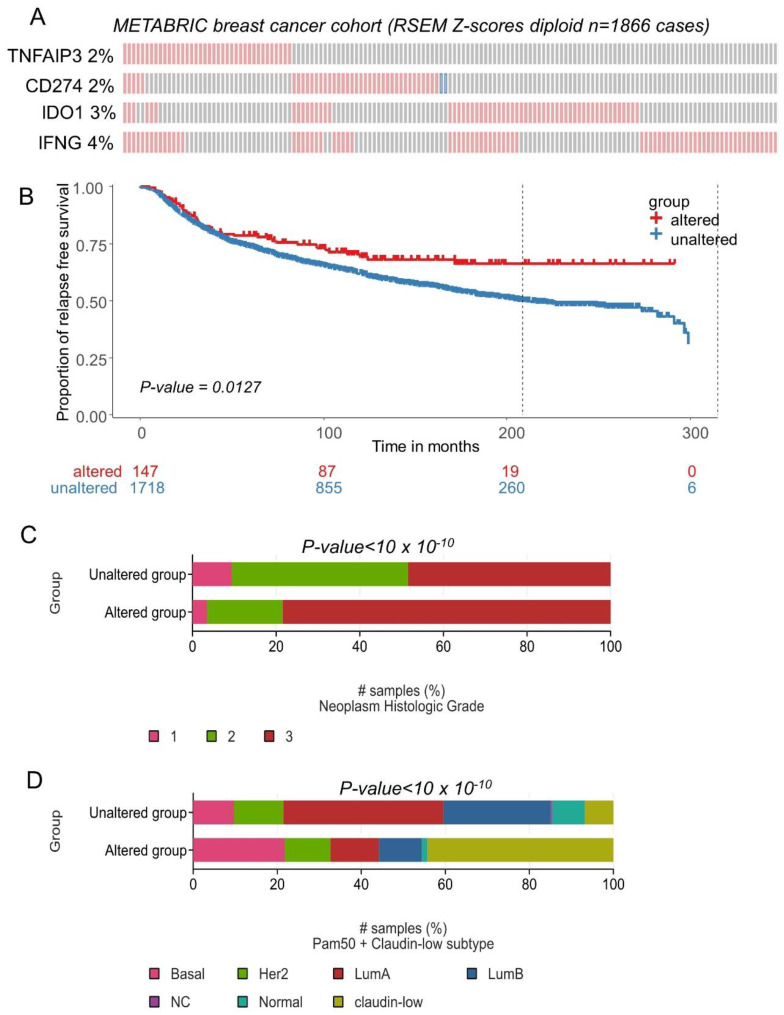
CD274 and ferroptosis drivers over expression in breast tumors was confirmed in patients with longer relapse free survival (METABRIC cohort): (**A**) Oncoprint of transcriptional alterations observed for CD274 and ferroptosis drivers (red: up expression, blue: down regulation, grey: unaltered); (**B**) Kaplan Meier and log rank test with relapse free survival as outcome and stratified on patient groups presenting or not alterations of CD274 and ferroptosis drivers in their transcriptome; (**C**) Barplot of Neoplasm histologic grade group proportions stratified on presence or absence of CD274-ferroptosis alterations; (**D**) Barplot of Neoplasm histologic grade group proportions stratified on presence or absence of CD274-ferroptosis alterations.

**Figure 9 cancers-16-00155-f009:**
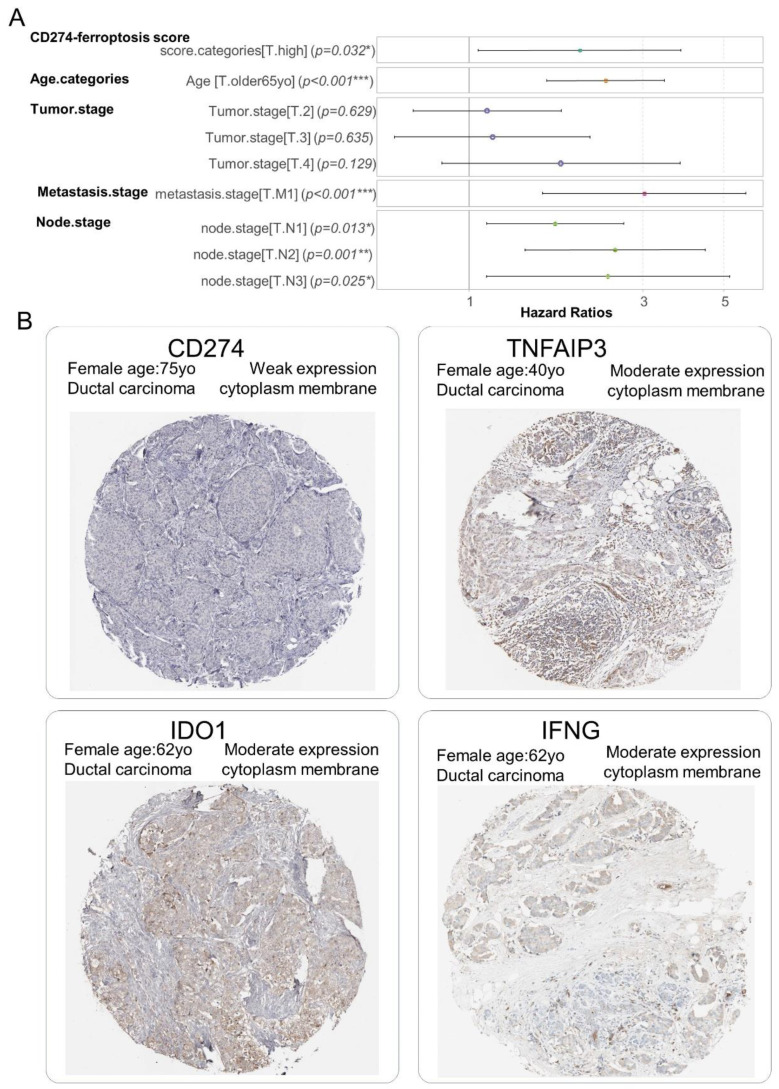
CD274 ferroptosis-driver score is independent prognostic marker in breast cancer: (**A**) Forest plot of multivariable Cox model with overall survival as outcome (TCGA cohort) and included CD274-ferroptosis score with other covariates such as age categories (threshold 65 yo), and TNM (Tumor, Nodes, Metastasis) classification, *: 0.01 ≤ *p* < 0.05, **: 0.001 ≤ *p* < 0.01, ***: *p* < 0.001, color points are representative of each covariate; (**B**) Representative detection at protein level (immunohistochemistry) in breast tumors for molecules included in computation of CD274-ferroptosis driver score: (low magnification).

**Table 1 cancers-16-00155-t001:** Univariate analyses on CD274 expression groups according to clinical parameters of patients in TCGA invasive breast cancer cohort.

Variable	Level	Low (*n* = 426)	High (*n* = 656)	Total (*n* = 1082)	*p*-Value
Age at diagnosis	Younger than 65 yo	283 (66.4)	462 (70.4)	745 (68.9)	
	Older than 65 yo	143 (33.6)	194 (29.6)	337 (31.1)	0.1871019
Tumor stage	T1	102 (24.1)	174 (26.5)	276 (25.6)	
	T2	239 (56.5)	388 (59.1)	627 (58.1)	
	T3	66 (15.6)	71 (10.8)	137 (12.7)	
	T4	16 (3.8)	23 (3.5)	39 (3.6)	0.1341858
	missing	3	0	3	
Node stage	N1	141 (33.9)	214 (33.1)	355 (33.4)	
	N0	196 (47.1)	316 (48.9)	512 (48.2)	
	N2	46 (11.1)	73 (11.3)	119 (11.2)	
	N3	33 (7.9)	43 (6.7)	76 (7.2)	0.8484744
	missing	10	10	20	
Metastasis stage	M0	336 (97.4)	558 (97.9)	894 (97.7)	
	M1	9 (2.6)	12 (2.1)	21 (2.3)	0.7909348
	missing	81	86	167	
Pam50 robust	LumB	154 (36.2)	169 (25.8)	323 (29.9)	
	Her2	31 (7.3)	80 (12.2)	111 (10.3)	
	LumA	166 (39.0)	245 (37.3)	411 (38.0)	
	Normal	14 (3.3)	28 (4.3)	42 (3.9)	
	Basal	61 (14.3)	134 (20.4)	195 (18.0)	0.0002416
Ggi score	mean (sd)	2.1 (12.6)	3.4 (13)	2.9 (12.9)	0.1112944
G70 score	mean (sd)	−0.23 (0.2)	−0.18 (0.2)	−0.2 (0.2)	0.0005523
Oncotypedx score	mean (sd)	61 (35.9)	65 (36.4)	63.5 (36.2)	0.0772518
OS_STATUS	Alive	356 (83.6)	575 (87.7)	931 (86.0)	
	Dead	70 (16.4)	81 (12.3)	151 (14.0)	0.0711620

**Table 2 cancers-16-00155-t002:** Univariate analyses of CD274 ferroptosis score categories according to clinical parameters of patients in TCGA invasive breast cancer cohort.

Variable	Level	Low (*n* = 153)	High (*n* = 929)	Total (*n* = 1082)	*p*-Value
Age at diagnosis	Younger than 65 yo	114 (74.5)	631 (67.9)	745 (68.9)	
	Older than 65 yo	39 (25.5)	298 (32.1)	337 (31.1)	0.124501
Tumor stage	T1	45 (29.4)	231 (24.9)	276 (25.6)	
	T2	93 (60.8)	534 (57.7)	627 (58.1)	
	T3	14 (9.2)	123 (13.3)	137 (12.7)	
	T4	1 (0.7)	38 (4.1)	39 (3.6)	0.061994
	missing	0	3	3	
Node stage	N1	46 (30.1)	309 (34.0)	355 (33.4)	
	N0	91 (59.5)	421 (46.3)	512 (48.2)	
	N2	12 (7.8)	107 (11.8)	119 (11.2)	
	N3	4 (2.6)	72 (7.9)	76 (7.2)	0.006523
	missing	0	20	20	
Metastasis stage	M0	140 (99.3)	754 (97.4)	894 (97.7)	
	M1	1 (0.7)	20 (2.6)	21 (2.3)	0.288441
	missing	12	155	167	
Pam50 robust	LumB	26 (17.0)	297 (32.0)	323 (29.9)	
	Her2	26 (17.0)	85 (9.1)	111 (10.3)	
	LumA	26 (17.0)	385 (41.4)	411 (38.0)	
	Normal	4 (2.6)	38 (4.1)	42 (3.9)	
	Basal	71 (46.4)	124 (13.3)	195 (18.0)	<0.0001
Ggi score	mean (sd)	10.5 (10.9)	1.6 (12.7)	2.9 (12.9)	<0.0001
G70 score	mean (sd)	0 (0.2)	−0.2 (0.2)	−0.2 (0.2)	<0.0001
Oncotypedx score	mean (sd)	84.8 (25.8)	59.9 (36.5)	63.5 (36.2)	<0.0001
OS_STATUS	Alive	141 (92.2)	790 (85.0)	931 (86.0)	
	Dead	12 (7.8)	139 (15.0)	151 (14.0)	0.025827

**Table 3 cancers-16-00155-t003:** Cox overall survival multivariate on TCGA invasive breast cancer cohort including CD274-ferroptosis score categories and clinical parameters.

Covariates	Hazard Ratios	CI95* Low	CI95* High	*p*-Value
CD274-ferroptosis score [T** high]	2.012	1.063	3.810	3.18 × 10^−2^
age categories [T** older than 65 yo]	2.368	1.635	3.430	5.13 × 10^−6^
tumor_grade [T** 2]	1.122	0.703	1.790	6.29 × 10^−1^
tumor_grade [T** 3]	1.161	0.627	2.149	6.35 × 10^−1^
tumor_grade [T** 4]	1.787	0.845	3.782	1.29 × 10^−1^
metastasis stage [T** M1]	3.021	1.594	5.726	6.98 × 10^−4^
node stage [T** N1]	1.726	1.123	2.653	1.28 × 10^−2^
node stage [T** N2]	2.517	1.426	4.445	1.46 × 10^−3^
node stage [T** N3]	2.405	1.118	5.174	2.48 × 10^−2^

CI95*: confidence interval at 95 percent; T**: reference.

## Data Availability

Recurrent scores for breast cancer computed on TCGA invasive breast cohort were shared as eset metadata of experiment assay in a R-package available online (tcga.breast): https://figshare.com/articles/software/R_package_composed_of_eset_RNAseq_-_TCGA_breast_cancer_with_clinical_scores_computed_/24083388/1; accepted date 22 December 2023, DOI: 10.6084/M9.FIGSHARE.24083388.V1. Specific R-packages designed for this article were deposited on github for a direct installation via devtools. It concerned “immunemod” package for immune modulation transcriptome study: https://github.com/cdesterke/immunemod, and ferroviz for ferroptosis transcriptome investigations: https://github.com/cdesterke/ferroviz.

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
