# Peer review of "Ferroptosis Inducers Upregulate PD-L1 in Recurrent Triple-Negative Breast Cancer"

_cancers, 2023, doi:10.3390/cancers16010155_

Round 1

Reviewer 1 Report

Comments and Suggestions for Authors

This investigation delves into the effects of ferroptosis inducers on both triple-negative breast cancer (TNBC) cells and breast tumors through a comprehensive transcriptome analysis. The findings highlight a notable upregulation of PD-L1 (CD274) expression in TNBC cells when exposed to ferroptosis inducers Erastin/RSL3, with a concomitant correlation to overall survival in breast cancer. Significantly, the study suggests a promising synergy between ferroptosis inducers and anti-PD-L1 immunotherapy, proposing an innovative therapeutic avenue for recurrent TNBC cases with heightened risk of relapse following neo adjuvant therapy. This discovery adds a compelling dimension to the current understanding of TNBC treatment strategies and holds potential clinical implications.

Major Comments:

1. Authors should indicate other immune modulators ( receptors) that are upregulated upon ferroptosis inhibitors.

2. What is the status of PD-1 when PD-L1 is upregulated? The author should monitor the protein level expression of these proteins.

3. Authors should try co-culture experiments to understand the effect of combination in the presence of immune system on cancer cells.

4. Results represented is from one cell line model. Too early to drive a conclusion from a single cell line. Author should use additional models of TNBC.

Author Response

REVIEWER 1

This investigation delves into the effects of ferroptosis inducers on both triple-negative breast cancer (TNBC) cells and breast tumors through a comprehensive transcriptome analysis. The findings highlight a notable upregulation of PD-L1 (CD274) expression in TNBC cells when exposed to ferroptosis inducers Erastin/RSL3, with a concomitant correlation to overall survival in breast cancer. Significantly, the study suggests a promising synergy between ferroptosis inducers and anti-PD-L1 immunotherapy, proposing an innovative therapeutic avenue for recurrent TNBC cases with heightened risk of relapse following neo adjuvant therapy. This discovery adds a compelling dimension to the current understanding of TNBC treatment strategies and holds potential clinical implications.

Major Comments:

  1. Authors should indicate other immune modulators (receptors) that are upregulated upon ferroptosis inhibitors.

Response to comment: Thank for your remark as you suggested, analyses of receptors from cancer immune landscape were also performed on MDA-MB-231 stimulated with erastin and RSL3 ferro-inducers. These analyses did not show any significant regulation of immune receptors. As these results did not show regulation they were added in a new supplementary figure and comments about these results were done in manuscript relative to the independence with PDL1 regulation. It makes sense with the new proteomic and single cell RNA-seq added to the manuscript: PDL1 is regulated during ferroptosis independently of its receptor PD1 (see next answers to your questions).

  1. What is the status of PD-1 when PD-L1 is upregulated? The author should monitor the protein level expression of these proteins.

Response to comment: We confirmed protein up-regulation of CD274 and TNFAIP3 induced by salinomycin (ferroptosis inducer) in breast CSCs model (human mammary epithelial HMLER CD24low/CD44high) by our spectrometric mass experiment published previously in Nature Cell Death and Disease 2023 (PMID: 37968262). PDCD1 alias PD1 was not found variable at protein level during these experimental conditions.

  1. Authors should try co-culture experiments to understand the effect of combination in the presence of immune system on cancer cells.

Response to comment: As data on coculture experiments did not exist for this type of stimulation (ferroptosis inducers on BC cell model with immune cells), we analyzed murine tumor microenvironment by single cell transcriptome (GSE191246) after transplantation of 4T1 stimulated or not by cyclophosphamide. Effectively, Cyclophosphamide-induced GPX4 degradation triggers parthanatos by activating AIFM1 (PMID: 35339754) and Cyclophosphamide induces the ferroptosis of tumor cells through Heme Oxygenase-1 (PMID: 35264971). These single cell transcriptome analyses confirmed induction of CD274 over-expression by cyclophophamide in 4T1 tumor immune microenvironment especially in B-cell, in Mo/Macro Lyz2+ or Skfn4+ and in dendritic cells Slamf7+. In 4T1 tumor microenvironment, TNFAIP3 appeared well expressed in T, B lymphoid and myeloid compartment but after cyclophophamide stimulation, this ferroptosis driver was found up regulated in: Mono/Macrophages skfn4+ and dendritic cells Slamf7+. In these experiments on 4T1 tumor microenvironment, Ido1 could not been detected. Pdcd1 (pd1) and Ifng were found invariably expressed in LyT Cd3e+-S100A10+.

  1. Results represented is from one cell line model. Too early to drive a conclusion from a single cell line. Author should use additional models of TNBC.

Response to comment: Results with ferroptosis inducers were presented in two distinct TNBC cell models: MDA-MB-231 and HCC38 in the initial manuscript. In the present manuscript, we confirmed PDL1 induction in immune micro-environment of 4T1 transplanted tumors by single cell transcriptome after ferroptosis induction with cyclophosphamide. CD274 induction at protein level was also confirmed by proteomic experiments with ferroptosis induction by salinomycin. During all these experiences, PDL1 inductions were observed independently of PD1 regulation.

Reviewer 2 Report

Comments and Suggestions for Authors

This article first sequenced the TNBC cell line and then analyzed it using two public databases to obtain genes related to ferroptosis and PD-L1 expression. Finally, it was verified using IHC method. The entire experimental design is sound, but please answer some questions:

1. There are classic pathways for ferroptosis, such as the p53 pathway and the non-p53 pathway. Please refer to the introduction section.

2. The LogFC parameters of the Limma package and the cut-off value of the p value should be reflected in the method section.

Author Response

REVIEWER 2

This article first sequenced the TNBC cell line and then analyzed it using two public databases to obtain genes related to ferroptosis and PD-L1 expression. Finally, it was verified using IHC method. The entire experimental design is sound, but please answer some questions:

  1. There are classic pathways for ferroptosis, such as the p53 pathway and the non-p53 pathway. Please refer to the introduction section.

Response to comment: Thank you for your remark. Effectively, TP53 is also very often mutated in triple negative breast cancer. Some comments leaking these ideas were added to the introduction part as your suggested.

  1. The LogFC parameters of the Limma package and the cut-off value of the p value should be reflected in the method section.

Response to comment: Thank you for your remark. These details were added to the corresponding material and method section: ‘Differential expression gene analysis between CD274-low and CD274-high patient samples was done with LIMMA R-package version 3.52.4 with thresholds on log2 fold change superior to 0.5 and on False Discovery Rate (FDR) adjusted p-value inferior to 0.05.’

Reviewer 3 Report

Comments and Suggestions for Authors

The paper is suitable for publication in Cancers. I just list a few clarifications:

1. in Materials and Metods website should be inserted as references;

2. in subsection 2.2.1 lines 110-114, check the statement structure, probably you should refer in separate periods about classifications and predictive scores;

3. in subsection 2.5 lines 148-150, use an equation to proprly write the score by introducing some symbols related with each expression;

4. in the first subsection of Results a pictorial scheme could be helpful in explaining a flowchart starting from clinical procedures;

5. in Figure 4D you should increase the fontsize.

Author Response

REVIEWER 3

The paper is suitable for publication in Cancers. I just list a few clarifications: Thank you very much for your comment.

  1. in Materials and Methods website should be inserted as references;

Response to comment: References were created and added for each cited websites especially for depository of software used during this study. See the new ‘Materials and Methods’ section.

  1. in subsection 2.2.1 lines 110-114, check the statement structure, probably you should refer in separate periods about classifications and predictive scores;

Response to comment: Thank you for your advice: effectively, the sentence was confusing and it was separate in two ones. One first sentence for classification predictions and one sentence for the scores computing.

  1. in subsection 2.5 lines 148-150, use an equation to properly write the score by introducing some symbols related with each expression;

Response to comment: A properly design equation was drawn with a formula software and added to the current manuscript.

  1. in the first subsection of Results a pictorial scheme could be helpful in explaining a flowchart starting from clinical procedures;

Response to comment: A flowchart with clinical procedure was added to the first subsection of the results (Figure 1).

  1. in Figure 4D you should increase the fontsize.

Response to comment: Thank you for your comment. The police size of the figure 4D was increased as much as possible for each phenotype items and gene names.

Round 2

Reviewer 1 Report

Comments and Suggestions for Authors

The authors have verbally addressed the comments. It would be great for the authors to run some experiments to validate the hypothesis.  

Author Response

Major Comments:

The authors have verbally addressed the comments. It would be great for the authors to run some experiments to validate the hypothesis.

Response to comment: Thank for your remark as you suggested, we ran some experiments to validate the hypothesis.

As indicated in the revised manuscript in page 9 (lines 300-305), upregulation of CD274 at surface protein level was confirmed by flow cytometry analyses in MDA-MB-231 (TNBC cell line) under treatment with erastin and RSL3 (shown in revised Figure 3E) but also in a breast cancer stem cell model: (human mammary epithelial HMLER CD24low/CD44high) under treatment with erastin, RSL3 and salinomycin (shown in revised Figure 4F).

Round 3

Reviewer 1 Report

Comments and Suggestions for Authors

The authors have addressed the comments.